

# Deep learning-based dimensional emotion recognition for conversational agent-based cognitive behavioral therapy

Julian Striegl[1], Jordan Wenzel Richter[2], Leoni Grossmann[3], Björn Bråstad[3], Marie Gotthardt[4], Christian Rück[3], John Wallert[3] and Claudia Loitsch[1]

[1] Center for Scalable Data Analytics and Artificial Intelligence (ScaDS.AI Dresden/Leipzig), Technische Universität Dresden, Dresden, Saxony, Germany
[2] Chair of Human-Computer Interaction, Technische Universität Dresden, Dresden, Saxony, Germany
[3] Centre for Psychiatry Research, Department of Clinical Neuroscience, Huddinge & Stockholm Health Care Services, Region Stockholm, Karolinska Institute, Stockholm, Sweden
[4] Kungliga Tekniska Högskolan, Stockholm, Sweden

Corresponding author
Julian Striegl,
julian.striegl@tu-dresden.de

## ABSTRACT

Internet-based cognitive behavioral therapy (iCBT) offers a scalable, cost-effective, accessible, and low-threshold form of psychotherapy. Recent advancements explored the use of conversational agents such as chatbots and voice assistants to enhance the delivery of iCBT. These agents can deliver iCBT-based exercises, recognize and track emotional states, assess therapy progress, convey empathy, and potentially predict long-term therapy outcome. However, existing systems predominantly utilize categorical approaches for emotional modeling, which can oversimplify the complexity of human emotional states. To address this, we developed a transformer-based model for dimensional text-based emotion recognition, fine-tuned with a novel, comprehensive dimensional emotion dataset comprising 75,503 samples. This model significantly outperforms existing state-of-the-art models in detecting the dimensions of valence, arousal, and dominance, achieving a Pearson correlation coefficient of $r = 0.90$, $r = 0.77$, and $r = 0.64$, respectively. Furthermore, a feasibility study involving 20 participants confirmed the model's technical effectiveness and its usability, acceptance, and empathic understanding in a conversational agent-based iCBT setting, marking a substantial improvement in personalized and effective therapy experiences.

## INTRODUCTION

Digitized therapeutic approaches have been researched for decades. Empirical studies yielded promising results regarding acceptance and efficacy for treatments of depression and anxiety disorders (*Etzelmueller et al., 2020*). Furthermore, examinations comparing internet-based cognitive behavioral therapy (iCBT) with traditional face-to-face therapy indicated comparable therapeutic effects on disorder symptoms associated with depression and anxiety (*Carlbring et al., 2018*). In recent years, scientific attention has been directed towards incorporating conversational agents (CAs) such as chatbots and voice assistants

into iCBT. While more clinical evidence is still needed, first studies indicate good effectiveness and acceptance by users (*Abd-Alrazaq et al., 2019*; *Milne-Ives et al., 2020*). CA-based CBT amalgamates the advantages of guided CBT with unguided therapeutic strategies (cf. impact of guidance on internet-based mental health interventions by *Baumeister et al. (2014)*), affording the capacity for adaptive interventions and personalized treatment approaches (*Mehta et al., 2021*). Moreover, CA-based CBT offers substantial scalability and represents a cost-effective possibility for therapeutic intervention. Consequently, CA-based CBT can be deployed as an autonomous therapeutic regimen or a supplementary component alongside conventional therapeutic modalities.

The exploration of one's emotions is a central component of CBT, as is a strong foundation of trust between the therapist and the individual conveyed through empathy. Therefore, therapeutic CAs must be capable of recognizing and tracking the emotions of patients to methodically respond to and empathize with them effectively in therapeutic conversations. Research substantiated a correlation between interpersonal factors (empathic understanding, positive regard, and congruence) and positive therapeutic outcomes (*Elliott et al., 2018*). It was observed that interpersonal factors have an influence on the effectiveness of therapeutic interventions that often surpasses the effect of the treatment method itself (*Lambert & Barley, 2001*). Therefore, established guidelines for the communication between therapist and patient should be considered when developing CAs for CBT. Furthermore, next to expressing empathy, emotional states can be used to assess intrinsic goals, exhibited behaviors (*Nelissen, Dijker & de Vries, 2007*), and treatment effect (*Hollon & Ponniah, 2010*), and successful emotion regulation within a session can be used as a potential predictor of long-term therapy outcome (*Mehta et al., 2021*). The impact of empathic conversations and emotion recognition is therefore an important area for research and development of CA in the field of CBT.

While there is already promising work in the field of CA-based CBT systems with emotion recognition capabilities (*Abd-Alrazaq et al., 2019*), established systems use a categorical emotional recognition approach thereby limiting the complexity of tracked emotional states to predefined categories instead of the finer-grained inference of a dimensional output vector (*Gabriels, 2019*). Approaches for deep learning-based dimensional text-based emotion recognition have been proposed but thus far have not been used in the context of CA-based CBT and, moreover, could be improved upon by means of dataset merging and fine-tuning thereby addressing core challenges in emotion recognition such as heterogeneous annotation methods and domain transfer (*Al Maruf et al., 2024*). Furthermore, while numerous studies have looked at the general acceptance and user satisfaction of CA-based CBT systems as a whole, the isolated acceptance of emotion recognition in this context and the perceived empathic understanding have thus far not been investigated. We address this limitation through the following contributions:

1. Development of a transformer-based model: We introduce a transformer-based model for dimensional text-based emotion recognition that captures a broad spectrum of emotional complexities. This model significantly outperforms existing state-of-the-art models in recognizing emotional states' dimensions, specifically valence, arousal, and dominance (VAD), based on individual text messages.

2. Creation of a new dimensional emotion dataset: We created a novel dimensional emotion dataset using a dataset transformation schema that integrated both publicly available categorical and dimensional data sources into one data pool. This new dataset, comprising 75,503 samples, features a more balanced distribution of VAD and was crucial for fine-tuning our model.

3. Application to CA-based CBT: The advanced emotion recognition capabilities of our model were integrated into a CA-based CBT system. A feasibility study involving 20 participants evaluated not only the model's technical effectiveness but also its usability, acceptance, and ability to convey empathy, which are key factors for applications in therapeutic contexts such as iCBT.

The central innovation of our work is the interdisciplinary approach that combines advancements in artificial intelligence (AI), particularly in emotion recognition, with applied computer science and research in internet-delivered psychotherapy to enhance iCBT. This integration allows for more precise and empathic understanding of patients' emotional states, which could lead to significant improvements in therapy outcomes, *e.g.*, in tracking emotional changes over time, assessing therapy progress, and potentially predicting long-term outcomes.

## RELATED WORK

Woebot (WoeBot Inc., https://woebothealth.com/, accessed 27.07.2023) is a mental health application that aims to provide content and techniques based on CBT to users *via* an integrated chatbot. Users can track their emotional states by selecting predefined emotional categories and by categorizing their moods through questions and answers. The tracking of emotional states, however, uses a categorical emotional model, thereby limiting the complexity of collected emotions (*Gabriels, 2019*). *Fitzpatrick, Darcy & Vierhile (2017)* investigated the acceptance and efficacy of the system with 70 participants in a randomized controlled study over a two-week period. Their results suggest a good acceptance and showed a significant decrease in symptoms of depression and anxiety in comparison to an e-book information control group. The acceptance and accuracy of the mood tracking functionality in the application have thus far not been investigated separately.

Youper (https://www.youper.ai/, accessed 27.07.2023) is a chatbot-based system providing exercises and content based on CBT to help people deal with emotional distress *via* emotion regulation approaches in an in-time intervention approach. Users can use a daily check-in functionality to track basic emotional states in a discrete categorical approach, combined with the possibility of recording an intensity level for the chosen emotional category. The acceptance and effect of the system on emotion regulation capabilities and symptoms of depression and anxiety were investigated in a longitudinal observational study with active customers of the platform by *Mehta et al. (2021)*. Results indicate a good acceptance and a decrease in depression and anxiety symptoms after two weeks of usage. The acceptance was measured through a 5-star rating. Standardized questionnaires for determining acceptance, such as the Client Satisfaction Questionnaire for web-based health interventions (CSQi) (*Boßet al., 2016*) or the Net Promoter Score (NPS) (*Baehre et*

*al., 2022*), were not taken into account. This impedes the assessment of the comparability of the study results with other systems.

The Wysa (Wysa, https://www.wysa.com/, accessed 27.07.2023) system combines chatbot-based therapeutic exercises with human mental health coaching. In the application, users can track their mood by choosing a predefined emotional category at the beginning of each session with the system. The application was used in multiple prospective cohort studies with participants with symptoms of anxiety and depression (*Leo et al., 2022*; *Inkster, Sarda & Subramanian, 2018*). Results showed high patient engagement and improvements in anxiety and depression scores among participants of the high-usage group when compared to the low-app-usage group. However, a separate evaluation of mood tracking is also missing in this study.

Cass AI (X2AI, https://www.cass.ai/x2ai-home, accessed 28.07.2023) is a system designed to deliver adaptive conversations based on expressed emotions and mental health concerns of users (*Joerin, Rauws & Ackerman, 2019*). Users can interact with the system either *via* free text input or by selecting predefined answers. According to the company, the system can detect patterns in phrasing, typing length of sentences, and number of grammatical errors to reveal dependencies to different emotional categories. The system was used by *Fulmer et al. (2018)* in a study to investigate its efficacy in reducing symptoms of depression and anxiety in college students. The authors conducted a single-blind randomized user study with an information control group (receiving an educational e-book on the topic). Results showed a significant reduction in symptoms of anxiety and depression in the experimental group. Participants from the experimental group showed higher levels of engagement and user satisfaction than those from the control. The accuracy of detected emotions and the level of conveyed empathy of the system have thus far not been investigated.

In contrast to applying categorical emotion recognition approaches in the context of CA-based CBT, there is no comparable research on using dimensional text-based emotion recognition in this context. However, there is some work on dimensional text-based approaches to emotion recognition in other contexts, which will be discussed subsequently.

*Al Maruf et al. (2024)* conducted a survey on the current state-of-the-art, challenges and opportunities of text-based emotion detection. The authors identified suitable datasets and libraries for model training and created a meticulous overview of approaches for text-based emotion recognition including keyword and lexical-based approaches, conventional machine learning, ensemble learning, and deep learning. The authors stated that most existing emotion classification systems use a categorical emotional model approach, thereby ignoring the complexity of emotional states. Furthermore, they found a constant unbalance in existing datasets resulting from a manual annotation process and mentioned that most datasets resulted from a specific domain which may impact model performance in different domains. In addition, they identified a lack of annotation standards, resulting in insufficient training data. Moreover, their results show heterogeneity in used evaluation metrics and in underlying datasets, thereby making the comparison of the performance of existing models difficult.

*Park et al. (2021)* fine-tuned a pre-trained Bidirectional Encoder Representations from Transformers (BERT) model (*Devlin et al., 2018*) to predict valence, arousal, and

dominance (VAD) scores from input sentences. As the number of available VAD mapping datasets is small, the authors transformed data from categorical emotion mappings (SemEval (SemEval task E-c, https://www.kaggle.com/datasets/context/semeval-2018-task-ec, accessed 22.08.2023) ISEAR (https://paperswithcode. com/dataset/isear, accessed 22.08.2023), and GoEmotions (*Demszky et al., 2020*)) using the NRC-VAD emotional dictionary (*Mohammad & Turney, 2010*). Results outperformed the previous state-of-the-art in accuracy ratings for each VAD value. Their results demonstrate the feasibility of the mapping approach from categorical to dimensional emotional data and the potential of fine-tuned BERT models for emotion recognition. However. the authors did not pool data sets for training and did not investigate the acceptance and feasibility of the model in the context of CA-based CBT.

*Yang et al. (2023)* employed the VAD affect representation for emotion recognition in conversations using cluster-level contrastive learning. In a similar approach, they used disentangled variational autoencoders (*Yang, Zhang & Ananiadou, 2023*) for VAD-based emotion recognition based on conversation histories. With both models, new state-of-the-art results could be achieved for two datasets. However, their approach is only applicable if a conversation history is already available. This cannot be a precondition for use in the area of CA-based CBT, because ethical considerations mean that context-free individual messages are preferable.

*Ghafoor et al. (2023)* developed a context-aware emotion classifier called *TERMS* using a Gaussian mixture model for soft assigning emotional perceptions in a valence-arousal space. The classifier maps emotional perspectives as distributions into a multidimensional space based on single texts. The authors evaluated their model on 4,000 text messages from the platform Twitter that were annotated with valence and arousal scores using Amazon Mechanical Turk. They compared the performance of their system to baseline classifiers, including naive Bayes, support vector machines, gradient boosting, and convolutional neural networks, and to state-of-the-art models, including DeepMoji (*Felbo et al., 2017*), SemEval-2018 Task 1 (*Baziotis et al., 2018*), a variational autoencoder-based model (*Wu et al., 2019*), and a context-aware emotional classifier deep learning model (*Song, Petrak & Roberts, 2018*). Their model outperformed the past state-of-the-art with a Pearson correlation coefficient ($r$) of $r = 0.6$ for valence and reached comparative results to the SemEval-2018 Task 1 model for arousal ($r = 0.3$). The authors furthermore investigated and noted the effects of different annotation methodologies on model performance.

In summary, it can be concluded that established CA-based CBT systems offer mood tracking features, but thus far use a categorical emotion recognition approach, which limits the complexity of the recorded emotional states to predefined categories. In addition, no isolated evaluations of emotion recognition have been carried out in studies of these systems, which means that the accuracy and acceptance by users cannot be assessed individually. Furthermore, the state-of-the-art reveals that approaches for deep learning-based dimensional text-based emotion recognition have previously been proposed but have not yet been used and researched in the application area of CA-based CBT. As we will demonstrate in this article, existing approaches could moreover be enhanced by pooling

different datasets to generate a more balanced and diverse data basis for fine-tuning a deep learning-based model.

# DIMENSIONAL EMOTION RECOGNITION FOR CA-BASED CBT

Our objective is to enhance iCBT through fine-grained dimensional emotion recognition, aiming to improve empathic understanding of patient's emotional states. Accordingly, we researched and developed an emotion-sensitive CA-based CBT system capable of performing text-based dimensional emotion recognition on individual user utterances. The system is detailed in the following subsections, with particular attention to three key contributions: the transformer-based model ('Transformer-based Model for Dimensional Emotion Recognition'); the creation of a new dimensional emotion dataset ('Emotion Dataset Merging and Training'); and the application to the CA-based CBT context ('Overall Conceptual Design' and 'Implementation').

## Overall conceptual design

The emotion-sensitive CA-based CBT system is powered by a dialogue management component that comprises both textual (chatbot (CB)) and vocal (voice assistant (VA)) means of communication to ensure accessibility for as many people as possible, including people with disabilities. The concept of the emotion-sensitive CA is illustrated in Fig. 1. For the VA, user input is first transcribed by a speech recognition component and converted into text. The dialogue management system then infers user intent, recognizes the emotional state of the user, and returns the respective system response (see Fig. 1A). While the structure of the CA-based CBT session itself is static (*i.e.,* the system responses are not generated but are predefined in a closed-domain dialogue approach) and can consist of several CBT-based exercises, custom empathic responses are delivered at certain system states to simulate therapist empathy by inferring upon the emotion and selecting from a set of manually tailored empathic responses in a decision-tree approach. Finally, the textual response is synthesized, transformed back to vocal data, and returned to the user.

To demonstrate the benefits of empathic system responses in the context of CBT and to evaluate its acceptance, we propose a modular CBT session, which can be flexibly adapted and augmented (see Fig. 1B). First, users are introduced to the system and familiarized with the input modalities, session contents, requirements, and duration. Subsequently, the user's suicidal tendencies are assessed, with a referral to human-operated suicidal hotlines if confirmed. Following these preliminaries, a sequence of emphatically enhanced CBT modules can be arranged tailored to the patient's needs. An exemplary mood monitoring module is proposed with a range of emotion-inquisitive questions to leverage the empathic capabilities of the CA as much as possible. The questions are designed to elicit open and complex input from the user regarding their feelings and experiences in the past, present, and future without limiting themselves to basic emotional categories.

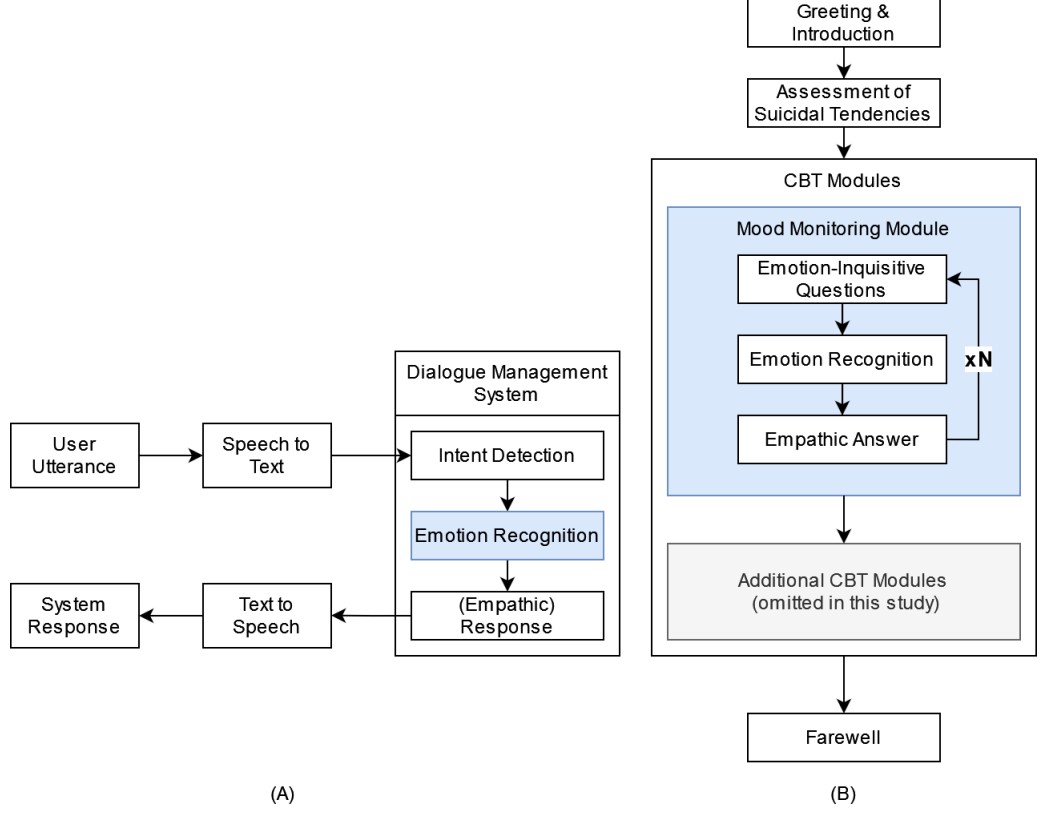

**Figure 1** **Conceptual design of emotion-sensitive CA-based CBT.** Components of an emotion-sensitive CA are depicted on the left (A): The emotional state and intent are detected based on provided user input (user utterance) and an empathic answer is returned to the user. The structure of an empathic CA-based CBT session is depcited on the right (B).

## Transformer-based model for dimensional emotion recognition

The focus of this research is on the emotion recognition component of the conceptualized CA-based CBT system described in the previous section. Therefore, this section focuses on the development of the DL-based approach for dimensional emotion recognition from text. The DL-based approach utilizes a BERT architecture (*Devlin et al., 2018*) with an added final regression layer for computing a dimensional output (Fig. 2). Specifically, a pre-trained ALBERT model (*Lan et al., 2019*) is used, which represents each word and sub-word unit as a vector in a higher dimensional space, taking into account the surrounding context (non-linear mapping), is fine-tuned on emotion-annotated data, and learns to infer a dimensional score on input sentences in an added linear regression layer. This linear regression layer models the relationship between the textual input data and the dimensional emotion label.

Hyperparameter tuning and choosing of an appropriate ALBERT backbone were done using a train/validation/test split of 78.4%, 19.6%, and 2%, resulting in 59,032, 14,759, and 1,506 samples respectively. The rather small test split still resulted in a sizeable sample size and was deemed sufficient for detecting overfitting of the validation set

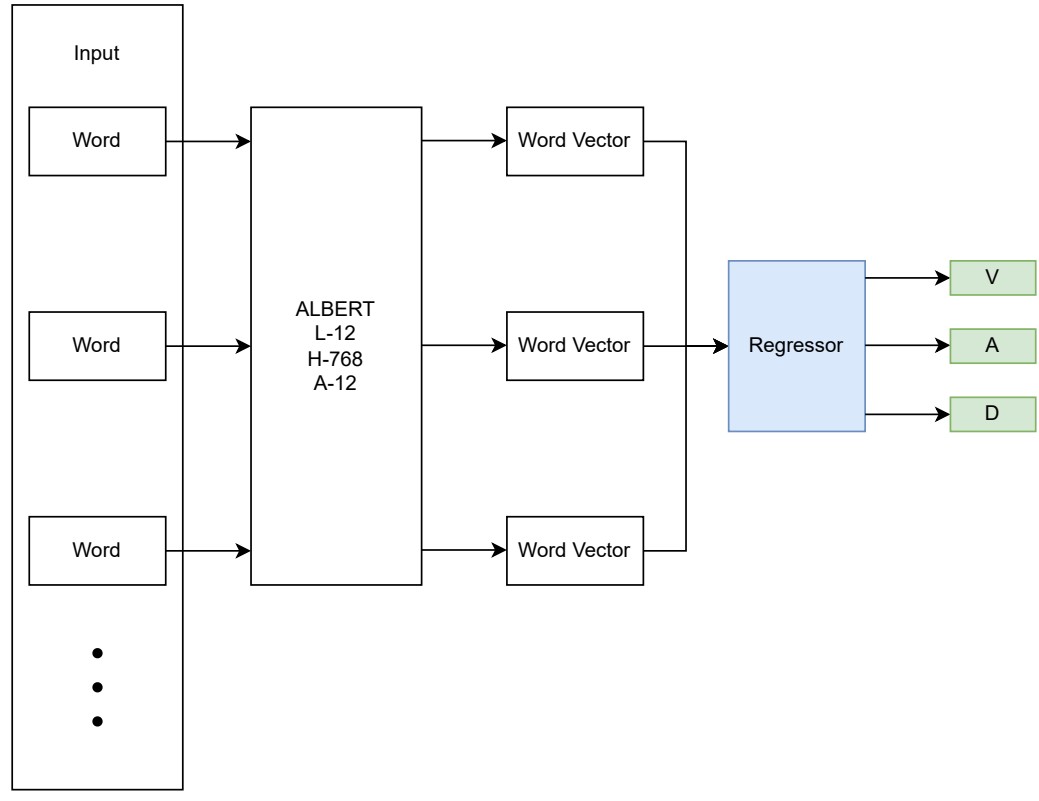

**Figure 2** DL-based emotion recognition module.

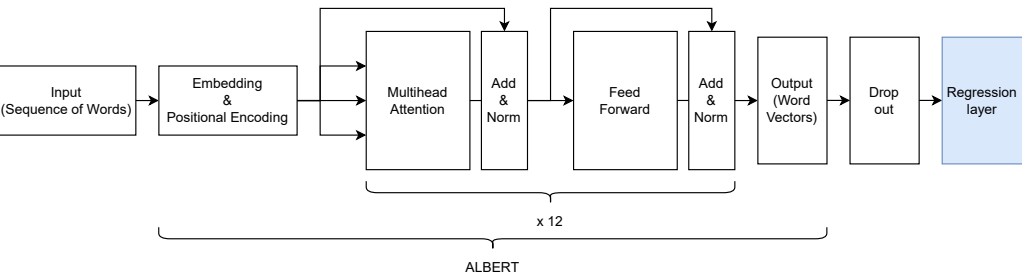

**Figure 3** Architecture of the created ALBERT-based model for dimensional emotion recognition.

during hyperparameter tuning. The final model architecture (see Fig. 3) comprised an input layer with variable input size, a preprocessing layer, a pre-trained ALBERT encoder (https://www.kaggle.com/models/ tensorflow/albert/frameworks/tensorFlow2/ variations/en-base/versions/2, accessed 22.11.2023.) with 12 hidden layers/transformer blocks having the size of 768 units with GeLu activation and 12 attention heads, a dropout layer with dropout value 0.1 to prevent overfitting and a final dense layer with linear activation for predicting into the 3 VAD classes.

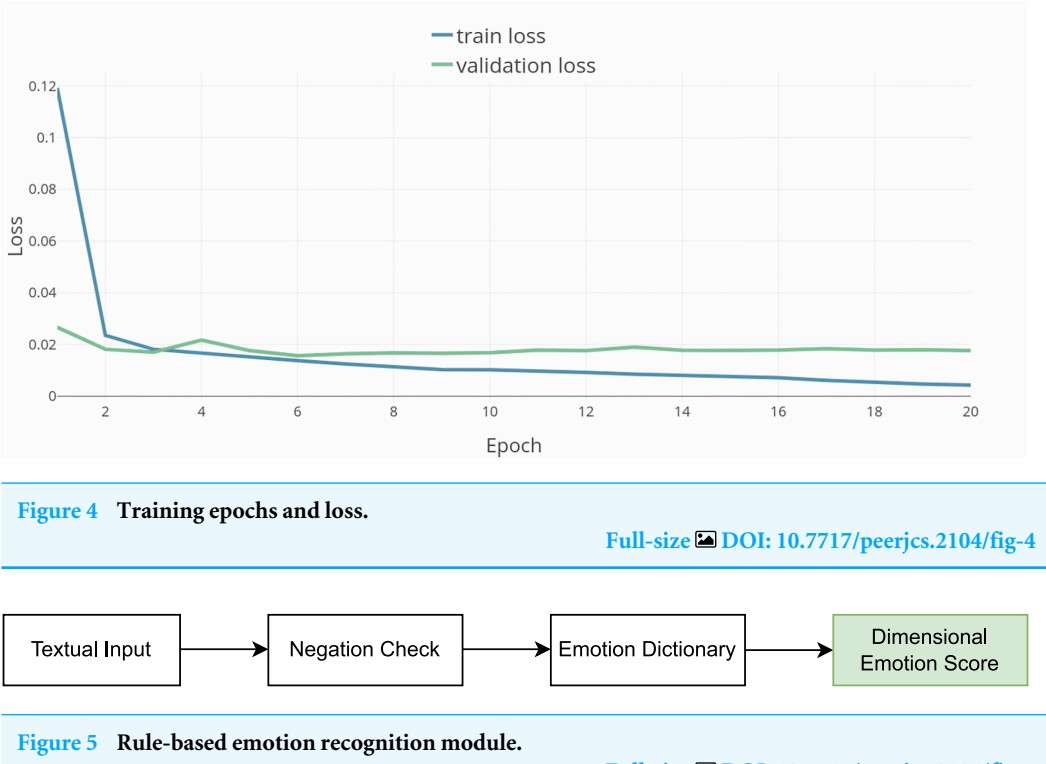

**Figure 4** Training epochs and loss.

Textual Input → Negation Check → Emotion Dictionary → Dimensional Emotion Score

**Figure 5** Rule-based emotion recognition module.

The encoder was trained on text from the English Wikipedia (Wikipedia Corpus, https://huggingface.co/datasets/wikipedia, accessed 23.03.24) and Book corpus (https://huggingface.co/datasets/bookcorpus, accessed 23.03.24), sizing 25 GB altogether. In total, the model has 16 layers with 11,685,891 trainable parameters. The mean squared error was chosen as a reliable loss function for regression tasks, an adaptive learning rate of 3e−5, AdamW as the optimizer for penalizing large weights, and a batch size of 32.

Training was done in 20 epochs (see Fig. 4). The resulting model and dataset were published open-source under MIT license (Model repository, https://github.com/JulianStriegl/dimensional-er-cbt, accessed 02.04.24).

To evaluate the DL-based emotion recognition approach in a comparative study ('Evaluation') we further developed an auxiliary rule-based approach based on *Badugu & Suhasini (2017)*. It first checks for negations before using the dimensional emotion dictionary by *Kušen et al. (2017)* to look up the emotion score associated with each word of the input and aggregate the results into a final emotional score (Fig. 5).

## Emotion dataset merging and training

To create the best possible basis for fine-tuning the created model, available datasets were combined into a large dataset pool. This was done to increase the amount of samples for training and to address key challenges in the field of text-based emotion recognition (cf. *Al Maruf et al., 2024*) by creating a more balanced dataset that covers different domains and annotation types, has a good balance between the different VAD dimensions, and thus leads to good generalizability. As most available emotion

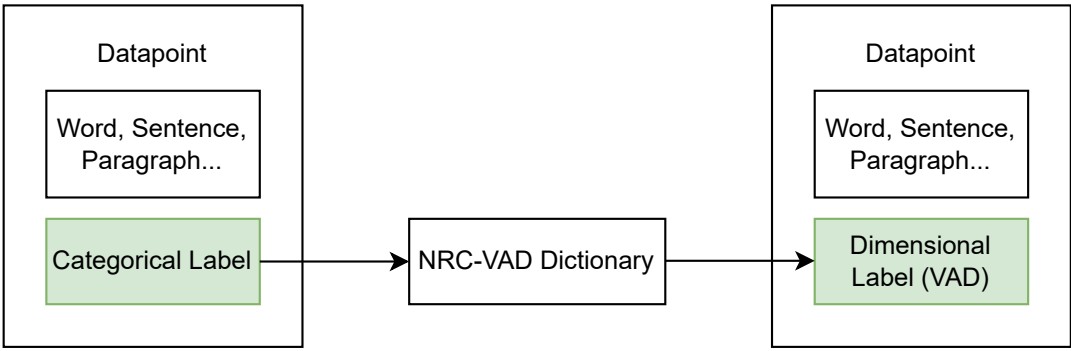

**Figure 6** Transformation scheme from categorical to dimensional datasets using a emotion dictionary.

**Table 1** Comparison of datasets in regards to the dataset size and label type (native VAD labels, categorical labels).

| Dataset | Size | Label type |
| --- | --- | --- |
| EmoBank | 10,000 | Native VAD labels |
| GoEmotions | 58,000 | Categorical |
| ISEAR | 7,503 | Categorical |
| CrowdFlower | 7,500 | Categorical |

datasets are categorically labeled, a dataset transformation scheme was used (see Fig. 6). Categorical labels in a given dataset were transformed into dimensional labels by taking the corresponding score in the NRC-VAD (National Research Council Canada, https://saifmohammad.com/WebPages/nrcvad.html, access date 13.08.2023) a dimensional emotion dictionary. The mean of the score was computed for multiple labels.

Four suitable datasets, namely EmoBank, GoEmotions, ISEAR, and CrowdFlower were identified and transformed if they were not inherently dimensionally annotated (see Table 1). The distribution of valence, arousal, and dominance in the individual datasets after transformation, as well as in the combined dataset can be seen in Fig. 7.

- The Emobank dataset (*Buechel & Hahn, 2022*) consists of over 10,000 annotated English sentences, sourced from previously categorically annotated datasets (the manually annotated sub-corpus of the American National Corpus (*Ide et al., 2008*) and the SemEval-2007 Task 14 AffectiveText Corpus (*Strapparava & Mihalcea, 2007*). Each sentence in the dataset was rated on its VAD value by five distinct human judges.
- The GoEmotions dataset (*Demszky et al., 2020*) consists of 58,000 annotated comments of the social media platform reddit.com (https://www.reddit.com/, accessed 23.11.2023), making it the largest emotion dataset annotated by humans currently. Each sentence in the dataset was labeled by three, and in cases of indecision, five human judges with one of 28 emotional categories.
- The International Survey on Emotion Antecedents and Reactions (ISEAR) (ISEAR dataset, https://paperswithcode.com/dataset/isear, accessed 22.08.2023) is an annotated

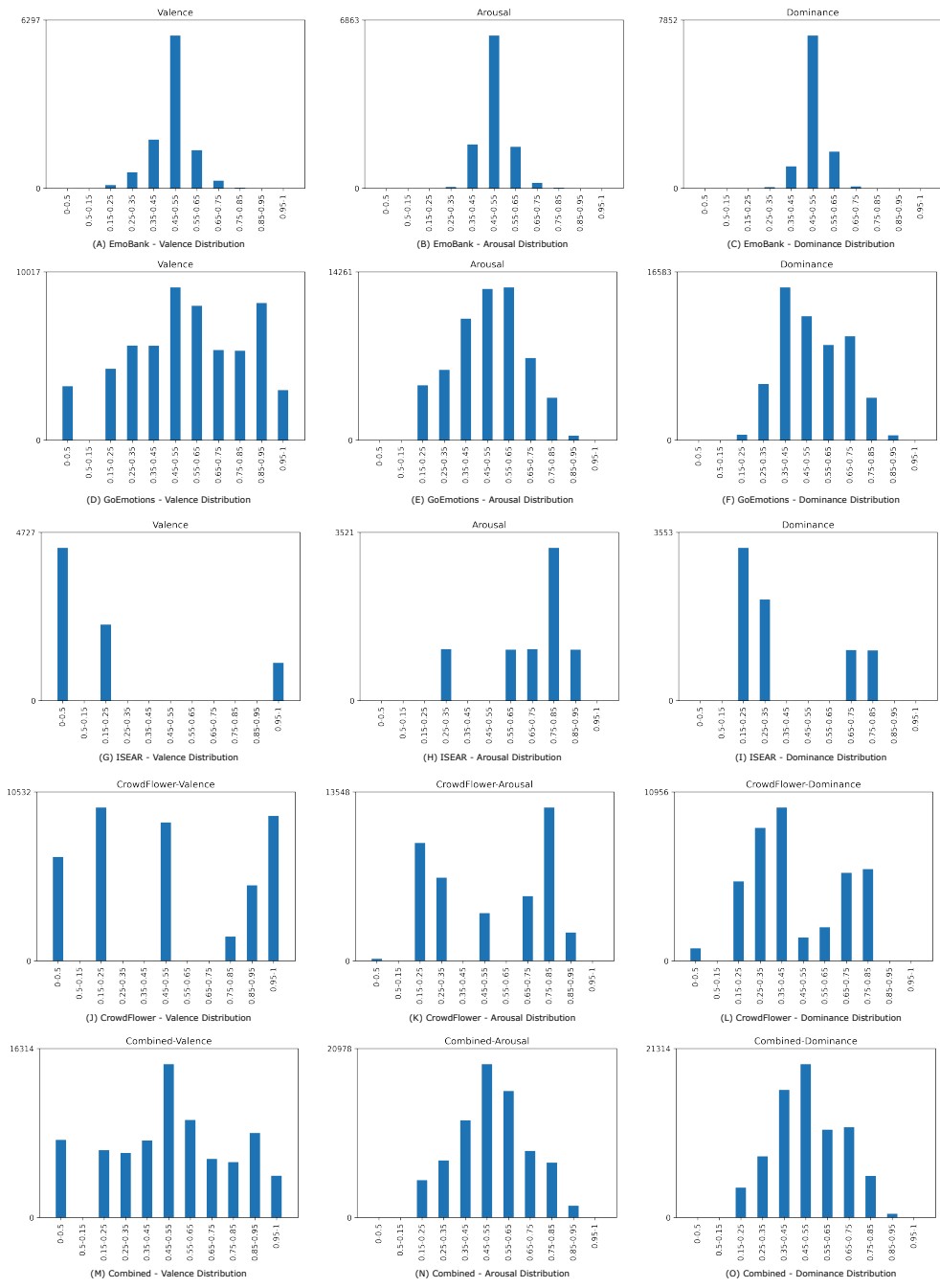

**Figure 7  Data distribution for valence, arousal, and dominance for the underlying and the combined dataset.**

dataset that comprises 7,503 sentences, annotated for the emotional categories of joy, fear, anger, sadness, disgust, shame, and guilt by psychology students and non-psychology students.

- The CrowdFlower dataset (Sentiment Analysis in Text - Dataset by crowdflower, https://data.world/crowdflower/sentiment-analysis-in-text, accessed 07.08.2022) is comprised of over 7,500 tweets, annotated by the emotional categories: empty, sadness, enthusiasm, neutral, worry, sadness, love, fun, hate, happiness, relief, boredom, surprise or anger.

The CrowdFlower dataset was exempt from training and reserved as an independent evaluation dataset to investigate the generalization ability of the trained model for different labeling heuristics. The dataset was chosen for this as it has an uneven spread of emotional categories, making it more suitable for testing than training data. The remaining datasets were merged to create one large dataset. As shown in Fig. 7, combining EmoBank, GoEmotions, and ISEAR yielded a dimensional emotion dataset with 75,503 samples and a more balanced VAD distribution than their individual constituents. Furthermore, dataset merging addresses the bias of heterogeneous manual annotation approaches and combines data from multiple domains, thereby potentially improving model generalizability and performance (cf. *Al Maruf et al. (2024)*).

## Implementation

To use the trained model in the context of CA-based empathic CBT, a web application was implemented based on the previously developed concept (see 'Overall Conceptual Design'). The user interacts with the CA *via* voice input or chat through a web-based user interface. The conversation with the CA is represented by speech bubbles to resemble a conversation with a human. A record button and an alternative text input field below the conversation can be utilized by the user to record answers to the CA. Open-source solutions were utilized for the web server, dialogue management, speech recognition, and speech synthesis, namely Flask (https://flask.palletsprojects.com/en/3.0.x/, accessed 23.10.2023), RASA (https://rasa.com/, accessed 23.10.2023), Vosk (https://alphacephei.com/vosk/, accessed 23.10.2023), and Rhasspy (https://rhasspy.readthedocs.io/en/latest/, accessed 23.10.2023).

## EVALUATION

For evaluation, the chosen performance metrics of the fine-tuned model were compared to the alternative rule-based approach and to related work in the field. Additionally, a user study was conducted to assess the acceptance of the performed emotion recognition, empathic understanding capabilities of the system, and perceived appropriateness of empathic system responses in the context of CA-based CBT.

### Technical evaluation

In the following section, the quantitative results and applied methodology of the technical evaluation will be described.

#### *Methodology*

Mean squared error (MSE) and Pearson correlation coefficient ($r$) were measured for both the DL-based and rule-based approach for comparative purposes. As the DL-based approach has been trained on most of the datasets, *Split* evaluates only the parts of the

**Table 2** Mean squared error for valence, arousal, dominance, and combined VAD-score for DL- and rule-based emotion recognition approach on different datasets.

| | Mean squared error | | | | | | | |
|---|---|---|---|---|---|---|---|---|
| | DL-Group | | | | Rule-Group | | | |
| | V | A | D | VAD | V | A | D | VAD |
| EmoBank | 0.0019 | 0.0019 | 0.0017 | 0.0014 | 0.0306 | 0.0221 | 0.0180 | 0.0198 |
| GoEmotion | 0.0042 | 0.0041 | 0.0029 | 0.0029 | 0.0532 | 0.0358 | 0.0270 | 0.0319 |
| ISEAR | 0.0011 | 0.0013 | 0.0011 | 0.0008 | 0.2125 | 0.0974 | 0.0734 | 0.1104 |
| Combined | 0.0036 | 0.0036 | 0.0026 | 0.0025 | 0.0661 | 0.0401 | 0.0304 | 0.0381 |
| Split | 0.0015 | 0.0017 | 0.0015 | 0.0011 | 0.2066 | 0.1014 | 0.0731 | 0.1098 |
| CrowdFlower | 0.0935 | 0.0765 | 0.0470 | 0.0614 | 0.1249 | 0.0889 | 0.0603 | 0.0793 |

**Table 3** Pearson correlation coefficient for valence, arousal, dominance, and combined VAD-score for deep learning- and rule-based emotion recognition approach on different datasets.

| | Correlation coefficient $r$ | | | | | | | |
|---|---|---|---|---|---|---|---|---|
| | DL-Group | | | | Rule-Group | | | |
| | V | A | D | VAD | V | A | D | VAD |
| EmoBank | 0.8952 | 0.7674 | 0.6436 | 0.7687 | 0.3627 | 0.1863 | 0.0749 | 0.2080 |
| GoEmotion | 0.9682 | 0.9285 | 0.9380 | 0.9449 | 0.4564 | 0.1826 | 0.2259 | 0.2883 |
| ISEAR | 0.9948 | 0.9831 | 0.9901 | 0.9893 | 0.3737 | 0.0224 | 0.2556 | 0.2172 |
| Combined | 0.9753 | 0.9401 | 0.9506 | 0.9553 | 0.4007 | 0.1608 | 0.1910 | 0.2508 |
| Split | 0.9925 | 0.9785 | 0.9861 | 0.9857 | 0.4118 | −0.0170 | 0.2562 | 0.2170 |
| CrowdFlower | 0.5333 | 0.2003 | 0.4624 | 0.3987 | 0.2968 | 0.1047 | 0.1712 | 0.1909 |

datasets that have not been used for training. Additionally, the DL-based and rule-based approach were compared in regard to Micro, Macro, and Average F1 score. Furthermore, we compared the presented approach with the results from *Park et al. (2021)*, which used a similar dataset transformation scheme, but no dataset pooling, and with other related work in the field.

### Results

As shown in Tables 2 and 3, the DL model achieves smaller MSE and higher correlation throughout the datasets compared to the rule-based approach. It generally infers on the data more accurately, although as described earlier the annotation procedure of the different datasets varies.

To further compare the DL and rule-based approach, the recognized dimensional score was converted into categories. Five emotion words were chosen from the NRC-VAD, which were sufficiently equidistant both numerically and semantically, namely *empty*, *threatened*, *tranquil*, *excited* and *rooted*, with the valence-arousal scores of (0.188, 0.183), (0.052, 0.928), (0.917, 0.094), (0.908, 0.931) and (0.51, 0.527) respectively. Using these values as center points, five clusters in the valence-arousal space were created. Thus, the dimensional valence-arousal scores of the models could be interpreted into one of five distinct categories. Using this mapping scheme, it is possible to evaluate in terms of Precision, Recall, and F1

**Table 4** Individual precision, recall and F1 scores & combined F1 scores on the transformed outputs.

| | Combined dataset | | | | | |
|---|---|---|---|---|---|---|
| | **DL-Group** | | | **Rule-Group** | | |
| **Metric** | **Precision** | **Recall** | **F1** | **Precision** | **Recall** | **F1** |
| *Empty* | 0.87 | 0.66 | 0.75 | 0.26 | 0.04 | 0.07 |
| *Threatened* | 0.87 | 0.90 | 0.89 | 0.31 | 0.29 | 0.05 |
| *Tranquil* | 0.79 | 0.56 | 0.65 | 0.19 | 0.24 | 0.21 |
| *Excited* | 0.77 | 0.92 | 0.84 | 0.52 | 0.40 | 0.07 |
| *Rooted* | 0.89 | 0.90 | 0.89 | 0.61 | 0.94 | 0.74 |
| | Macro F1 | Micro F1 | Average F1 | Macro F1 | Micro F1 | Average F1 |
| *Total* | 0.80 | 0.86 | 0.86 | 0.23 | 0.59 | 0.48 |

**Table 5** Comparison of Pearson correlation coefficient results to results of *Park et al. (2021)* on EmoBank dataset.

| | Correlation coefficient *r* | | | | | | | | |
|---|---|---|---|---|---|---|---|---|---|
| | **Park et al.** | | | **DL-Group** | | | **Rule-Group** | | |
| | **V** | **A** | **D** | **V** | **A** | **D** | **V** | **A** | **D** |
| EmoBank | 0.84 | 0.57 | 0.52 | 0.90 | 0.77 | 0.64 | 0.36 | 0.19 | 0.07 |

**Table 6** Comparison of Pearson correlation coefficient of state-of-the-art models for recognition of valence and arousal based on the works of *Ghafoor et al. (2023)*. Results were obtained on different datasets.

| | Correlation coefficient *r* | |
|---|---|---|
| **Source/Model name** | **V** | **A** |
| *Felbo et al. (2017)*/DeepMoji | 0.54 | 0.23 |
| *Baziotis et al. (2018)*/SemEval-2018 | 0.59 | 0.40 |
| *Park et al. (2021)*/RoBERTa | 0.84 | 0.57 |
| *Ghafoor et al. (2023)*/TERMS | 0.60 | 0.30 |
| Our model (2024)/ALBERT | 0.90 | 0.77 |

on the combined dataset (as shown in Table 4). Analogous to the dimensional evaluation, the DL model outperforms the rule-based model on the combined dataset.

When comparing the Pearson correlation coefficient to the state-of-the-art DL model for dimensional text-based emotion recognition by *Park et al. (2021)*, the present model outperforms the one by Park et al. in all VAD dimensions (by $r = 0.06$ for valence, $r = 0.2$ for arousal, and $r = 0.12$ for dominance, see Table 5). Compared to the model by *Ghafoor et al. (2023)*, the presented approach outperforms for the dimension of valence by $r = 0.30$ and for arousal by $r = 0.47$ (see Table 6). The performance is, however, hard to compare with the model by Ghafoor and colleagues as they evaluated their model on a self-created unpublished dataset.

## User study

A randomized A/B-testing experiment was conducted as an online between-subject feasibility study with healthy individuals.

### Methodology

Participants were semi-randomly assigned to two groups. Hence, group assignment was done randomly while ensuring equal group sizes for maintaining balance and comparability between groups. Both groups were led through an exemplary CBT session, with the only difference being the DL-based emotion detection in one and rule-based emotion detection in the other group. Participants needed to sign a privacy policy and consent form to comply with data protection provisions.

Demographic information, symptoms of depression using the short version of the Patient Health Questionnaire (PHQ2) (*Löwe, Kroenke & Gräfe, 2005*), and the affinity of technical interaction (ATI) (*Franke, Attig & Wessel, 2019*) were measured as independent variables. Participants were asked to rate the perceived empathy, fluency, and relevance of system answers based on a 5-point Likert scale to assess the Empathic Understanding (EU) capabilities as proposed by *Rashkin et al. (2018)*.

The usability of the system was assessed using the System Usability Scale (SUS) (*Brooke, 1995*) as it is one of the most popular and validated instruments for usability assessment (*Bangor, Kortum & Miller, 2008*). The SUS investigates the perceived usability of a system with 10 questions based on a 5-point Likert scale, with the maximum score being 100 and a score above 68 being considered above-average usability.

The Client Satisfaction Questionnaire adapted to Internet-based Interventions (CSQi) (*Boß et al., 2016*) was used to investigate the acceptance of the system as it has been developed and validated specifically for digital mental health interventions. Each item of the CSQi is scored between 1 and 5. For determining the overall acceptance rating of the respective subject, scores are summed up, therefore ranging from 8 (lowest) to 32 (highest), with 20 being the medium score.

### Participants

Twenty participants (healthy individuals without a diagnosed mental health disorder) were recruited online and evenly split between the two groups (DL group and rule-based group). Age differences between the groups were not significant ($M_{DL} = 34.7$, $SD_{DL} = 12.45$, $M_{Rule} = 27.7$, $SD_{Rule} = 10.3$, $p = .21$). The differences between groups in terms of self-reported symptoms of depression (PHQ2) were not significant ($M_{DL} = 2.3$, $SD_{DL} = 1.85$, $M_{Rule} = 1.6$, $SD_{Rule} = 1.02$, $p = .33$). There were furthermore no significant differences in the measured technology affinity (ATI) ($M_{DL} = 32.2$, $SD_{DL} = 12.5$, $M_{Rule} = 39.2$, $SD_{Rule} = 8.68$, $p = .17$).

Half of the participants in the DL group and 40% in the rule-based group had prior experiences with VAs.

### Results

As shown in Table 7, questions concerning the participant's experience with the CA regarding EU showed no significant differences between groups in the combined score

**Table 7  User experience results for perceived empathic understanding (EU), system usability (SUS), and acceptance (CSQi) for the deep learning approach (DL Mean) and the rule-based control (Rule Mean) and standard deviation (SD).**

| Questionnaire | DL mean | DL SD | Rule mean | Rule SD |
|---|---|---|---|---|
| EU [3:15] | 10.50 | 02.25 | 10.90 | 01.87 |
| SUS [0:100] | 72.50 | 12.89 | 77.75 | 07.61 |
| CSQi [8:32] | 14.00 | 07.95 | 16.40 | 06.97 |

($M_{DL} = 10.5$, $SD_{DL} = 2.25$, $M_{Rule} = 10.9$, $SD_{Rule} = 1.87$, $p = 0.67$). The Rule-group scored higher in the *Empathy/Sympathy* ($M_{DL} = 2.9$, $M_{Rule} = 3.3$) and *Fluency* ($M_{DL} = 4.3$, $M_{Rule} = 4.7$) questions, whereas the DL-group scored higher in the question regarding *Relevance* ($M_{DL} = 3.3$, $M_{Rule} = 2.9$).

In the SUS ($M_{DL} = 72.5$, $SD_{DL} = 12.9$, $M_{Rule} = 77.8$, $SD_{Rule} = 7.62$, $p = .31$) and the CSQi ($M_{DL} = 72.5$, $SD_{DL} = 12.9$, $M_{Rule} = 77.8$, $SD_{Rule} = 7.62$, $p = .31$) no significant differences could be established. Altogether, no significant differences could be found between the experimental groups with the EU, SUS, and CSQi questionnaires. Both approaches achieved good usability and acceptance scores and scored high in empathic understanding.

# DISCUSSION

To investigate the feasibility of the proposed model in the context of CA-based CBT, this study reports on the technical evaluation as well as on a conducted feasibility study measuring the isolated acceptance and user satisfaction of the used emotion recognition, and perceived appropriateness of connected system responses. In the technical evaluation, the DL approach scored better than the rule-based approach in the metrics MSE, Pearson correlation coefficient, and F1 score. This finding was especially prominent when comparing the performance of data that were annotated under the same heuristics as it was trained on. When comparing performance on a foreign dataset, the CrowdFlower dataset, although the DL scored noticeably lower, it nevertheless outperformed the rule-based approach.

In comparison to related work, the here presented fine-tuned DL model outperformed the results of *Park et al. (2021)* in all three inferred dimensions. The main difference between the approaches being that Park and colleagues did not use a combined dataset for training and evaluated their model with a test set of EmoBank only. The performance gap between datasets with known and unknown heuristics underlines the importance of training prospective DL models on multiple datasets to promote a generalized understanding of emotions. Our model, furthermore, showed an improvement in Pearson correlation coefficient values for valence and arousal in comparison to the results of *Ghafoor et al. (2023)* and respectively to the models that Ghafoor and colleagues compared their model to. Their performance measurements were however performed on a self-created, unpublished dataset and are therefore hard to reproduce. This should be addressed in future research by comparing performance on the same test dataset.

In the user study, while no significant differences could be found between the two experimental groups regarding EU, SUS, or CSQi, the results indicate good usability and

acceptance for both groups. This is a rapidly growing research area, but it is still largely underdeveloped. Although it arguably has much potential, applied clinical research to date is scarce. Future research needs to explore the development and applications of these tools in clinical care. Although the results of the study demonstrate both the technical feasibility and the usability and acceptance by users in the context of CBT, further implications for use in the field of iCBT need to be considered, which are discussed subsequently.

## Implications of deep learning-based dimensional emotion recognition for iCBT

Implementation of AI tools in the mental healthcare context presents both opportunities and challenges. When assessing previous research on AI in mental health care, it is clear that there are flaws in research methodology and quality, such as not reporting external validation, high risk of bias, and a lack of transparency (_Tornero-Costa et al., 2023_). Ethical and legal issues come into play whenever automated machines are integrated in mental healthcare. When implementing AI in healthcare, _Gerke, Minssen & Cohen (2020)_ identified four primary ethical issues, which include,(1) informed consent to use, (2) safety and transparency, (3) algorithmic fairness and biases, and (4) data privacy, as well as five legal challenges: (1) safety and effectiveness, (2) liability, (3) data protection and privacy, (4) cybersecurity, and(5) intellectual property law. For AI to be used in the healthcare context, ethical considerations need to be made concerning informed consent to use. The aforementioned issues become more complex when they impact clinical decisions. Safety in the clinical context is a major concern regarding the implementation of AI, and it is important that developers prioritize reliable training data, transparency on the data used, and potential shortcomings of the system, _e.g._, biases. In order to avoid unfair outcomes, the data needs to be of high quality, unbiased, and therefore from a data science perspective balanced. Concerning data privacy, patients' trust through sufficient data protection measurements is crucial for the successful integration of AI in healthcare. When AI systems have access to patients' data, considerations need to be made concerning how and if patients are informed of the use.

Automation of time-consuming tasks in iCBT could, through a positive lens, lead to improved cost-effectiveness, which is an important point in often over-encumbered and underfinanced psychiatry treatment and care contexts. The WHO has identified digital health, which includes AI, as a critical step in the development of making healthcare more efficient and accessible worldwide (_World Health Organization, 2023_). Another potential benefit would be more consistency and thereby legal certainty on the patients' behalf. Even though contemporary iCBT consists of highly structured, randomized controlled trial-evaluated, treatment protocols that are executed by trained professionals, human therapists have limited memory and attention span, they will have "bad days", like one patient more than others, will have read one scientific paper but not another, possess differing skill and experience et cetera. All of the previously mentioned factors have the potential for varying the quality of the iCBT delivered. On the other side, variability in terms of human therapist performance may not matter that much in highly structured iCBT. Moreover, machines are not sentient beings and do not have a causal understanding of the

patient and his or her world. This means that the machine is susceptible to producing errors in assessment and treatment that are in part fundamentally different from human errors. iCBT treatment is a complex task and automating aspects of it is still in its infancy. The consequences of this paradigm shift are predominantly understudied to date. Considering the aforementioned, below follows a set of hypothetical scenarios of varying degrees of automation in the iCBT context focused on emotion recognition and CA treatment with respect to clinical implementation.

Fully automated iCBT, including the prediction of emotional states coupled with a CA in charge of the iCBT with no human therapist involvement, would be both unwanted and unethical. For legal reasons, having a clinical professional involved and ultimately responsible for treatment is mandatory today and unlikely to change in the foreseeable future. Only hybrid solutions of man-machine co-involvement are therefore further discussed here. One such hybrid scenario would be the sole automation of emotion recognition. This scenario starts with initial machine recognition of emotional states derived from patients' responses as part of ongoing iCBT treatment. Estimated emotional states can then be fed to a human clinician as decision support. In theory, this could render an improved understanding of a patient's emotional state and also change of state across time during iCBT. This could ultimately improve treatment tailoring and effectiveness through the patient perceiving the therapist as more empathic, strengthening the therapeutic alliance. Furthermore, it would allow for modifications of ongoing therapy work modules to better suit the patient's emotional state. A potential risk with this approach would be the drift of the therapist's own emotional assessment influenced by the machine's estimated emotional state of the patient which may be wrong or biased. An interesting but largely untested scenario would be extended automation of emotional recognition coupled with therapist-supported CA treatment. This would involve not only the potential benefit of emotion recognition discussed above but also cost-effective semi-automated treatment. One such implementation would be that the emotionally informed CA drafts empathically written therapy responses to the patient's messages and the human therapist then scrutinizes the responses and signs off on them with or without making prior changes. A major portion of iCBT costs come from therapists spending time drafting responses to patients in the treatment portal, unlocking a major potential for cost-saving strategies. An additional downside risk with this scenario would be that the human therapist—due to stress or other human factors—signs off on written responses of lower therapeutic quality. Proper training and structured follow-up of therapists are likely required in this scenario, which in turn may offset some of the cost-effectiveness of the approach. That stated since a major motivation for iCBT is cost-effectiveness, extending it with emotionally tailored CA seems in accordance with that overarching aim of iCBT.

## Limitations

The present study includes several limitations in terms of the system design choice, resource availability, and comparability.

### System design

First, the proposed system only imitated the beginning of a CBT session in order to investigate the isolated acceptance and appropriateness of recognized emotions and related empathic responses. Therefore, a follow-up study should investigate the acceptance and usability of a complete CA-based CBT session with an empathic agent using the presented approach for text-based emotion recognition. Second, empathic responses were given by choosing from five pre-defined answers based on the detected VAD score and the clustering approach described in 'Results'. While the recommendations of *Holtforth & Castonguay (2005)* and *Elliott et al. (2018)* concerning empathic responses in the therapy context were considered, no further investigation regarding the appropriateness of pre-defined answers was undertaken. While the sample size of 20 is too small to draw concise conclusions, this is a potential explanation for the relatively equal user study results for both experimental groups for SUS, CSQi, and EU. Future work should expand the system *e.g.*, by DL-based response generation to better leverage the advantages of dimensional emotion recognition in comparison to categorical approaches, making fine-grained tailored empathic answers possible.

### Emotion recognition approach and model training

Additionally, the presented approach focused on text-based emotion recognition of individual user utterances and did not take the history of the conversation into account. Therefore, based on the merged dataset a model for emotion recognition in conversations could be trained similar to the works of Yang and colleagues (*Yang et al., 2023*; *Yang, Zhang & Ananiadou, 2023*). Moreover, this work used text-based emotion recognition only (*e.g.*, for application in the context of chatbot-based CBT). Numerous research has investigated multimodal emotion recognition (cf. *Aslam et al., 2023*) and system performance might increase in an ensemble learning approach (as described by *Al Maruf et al. (2024)*) by combining the text-based emotion recognition with other modalities. The resulting model could be applied in the context of voice assistant-based CBT and embodied conversational agent-based CBT. The model could be, furthermore, tested on additional datasets, as the main test set in this study (CrowdFlower) had a varying labeling quality, thereby possibly influencing the results of the technical evaluation of the model. Regarding the training of the model, a relatively small parameter amount and training time was chosen, due to resource limitations. As the performance of pre-trained large language models usually increases with training time and network parameters used for fine-tuning (*Lan et al., 2019*), those factors should be increased for future versions of the system. Regarding the model architecture, ALBERT has been chosen as the backbone of the model architecture because of its training-efficient nature, but no further in-depth comparison of model alternatives has been undertaken, which could have revealed a more appropriate architecture for the task. As the field of research has the field of research on large language models is developing fast, the consideration of a suitable underlying model should be made constantly anew. Moreover, individual layers in the used architecture could be replaced in future iterations of the system to optimize performance. Furthermore, as the evaluation showed lower scores on test datasets with unknown heuristics, it can be hypothesized that training on

more heuristically distinct datasets could have increased the generalization ability of the model. Unfortunately, there are limited appropriate emotion-annotated datasets available. While we compared the model's performance to other state-of-the-art models, not all models and used datasets from related work are publicly available thereby impairing comparability. Furthermore, we mapped recognized emotional states to five emotional clusters to investigate the precision, recall, and F1 score of the implemented DL approach in comparison to the created rule-based approach. Future work could elaborate on this by mapping to basic emotional categories (*Ekman, 1992*) instead to enable a further comparison of the model's performance to categorical emotion recognition models.

### Biases in data and annotation

Though there are multiple raw data sources, together they still only originate from a handful of domains, each having its distinct slang and expressed emotions therein. It cannot be assumed, however, that participants of the user study shared the same domain-specific expressions, especially considering those participants whose first language was not English. Regarding the annotations, they too must be assumed to have inherent biases, due to individual differences in emotion understanding, varying annotator count, and different annotation methodologies. While merging multiple datasets into one larger dataset pool counteracts the effects of domain and annotation heterogeneity, effects are certainly still present. Furthermore, labeling into dimensional VAD scores is not intuitive, which might additionally have biased the annotation quality in the EmoBank dataset. Although the other datasets were categorically labeled, the transformation into dimensional scores with the NRC-VAD dimensional emotion dictionary likely inherited its existing biases. As more emotion-annotated training data becomes available, these biases in data and annotations can be addressed through a larger and more diverse dataset, likely resulting in a more robust and better generalizing model. Furthermore, it should be tested how the use of different emotion dictionaries for dataset transformation and merging affects model performance.

### Significance and comparability of user study

As we wanted to test the system's feasibility in a user study in an early stage of development, a small sample size of 20 participants was used thereby reducing the statistical significance and representativeness found results. Future research should address this limitation and evaluate in depth the perceived empathy and the use and impact of emotion recognition in the context of CA-based CBT with a larger and more diverse group of participants. As there have been thus far no other user studies investigating the isolated effect and acceptance of dimensional emotion recognition and empathic system responses in the context of CA-based CBT, there is difficulty in comparing found results with related work. As discussed earlier, related work up to now merely investigated the acceptance and effect of CA-based CBT systems as a whole thereby making causal inference regarding the effects of empathic dialogue management difficult.

## CONCLUSION

We presented a system for DL-based dimensional text-based emotion recognition for CA-based CBT. As a main contribution to the field, we developed a transformer-based model based on an ALBERT backbone, created a novel dimensional emotion dataset through a transformation and pooling process, and investigated the model's feasibility in the context of CA-based CBT in a feasibility study. In comparison to a rule-based approach, the presented system showed considerably higher scores in a technical evaluation and surpassed the state-of-the-art for text-based dimensional emotion recognition on individual user utterances. The conducted user study investigating the acceptance, usability, and empathic understanding of the developed system showed no significant differences between DL- and rule-based emotion recognition and connected empathic responses, probably due to the use of pre-defined answers. Results for both user groups showed good scores for usability, acceptance, and empathic understanding, thereby highlighting the model's feasibility to be used in the context of CA-based CBT. The presented model should be used in follow-up studies in combination with a more elaborate empathic response generation, a complete CA-based CBT session, and a larger sample size. Furthermore, the presented model could be improved through additional training datasets and a longer training time. Additionally, a different large language model could be used as a basis for fine-tuning, the effect of utilizing additional modalities could be investigated, and the conversation history could be taken into account to recognize emotional states.

### Funding
The authors received no funding for this work.

### Competing Interests
The authors declare there are no competing interests.

### Author Contributions
- Julian Striegl conceived and designed the experiments, analyzed the data, performed the computation work, prepared figures and/or tables, authored or reviewed drafts of the article, and approved the final draft.
- Jordan Wenzel Richter conceived and designed the experiments, performed the experiments, analyzed the data, performed the computation work, prepared figures and/or tables, authored or reviewed drafts of the article, and approved the final draft.
- Leoni Grossmann analyzed the data, authored or reviewed drafts of the article, and approved the final draft.
- Björn Bråstad analyzed the data, authored or reviewed drafts of the article, and approved the final draft.
- Marie Gotthardt analyzed the data, authored or reviewed drafts of the article, and approved the final draft.

- Christian Rück analyzed the data, authored or reviewed drafts of the article, and approved the final draft.
- John Wallert analyzed the data, authored or reviewed drafts of the article, and approved the final draft.
- Claudia Loitsch conceived and designed the experiments, analyzed the data, authored or reviewed drafts of the article, and approved the final draft.

### Data Availability

The model is available at GitHub and Zenodo:

- https://github.com/JulianStriegl/dimensional-er-cbt.
- Julian Striegl. (2024). JulianStriegl/dimensional-er-cbt: Initial Release (v1.0.0). Zenodo. https://doi.org/10.5281/zenodo.11091285.

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
