# Peer review of "Deep learning-based dimensional emotion recognition for conversational agent-based cognitive behavioral therapy"

_PeerJ Computer Science, doi:10.7717/peerj-cs.2104_

## Round 0.1 · original submission · Major Revisions

Dear authors,

Thank you for submitting your paper. The reviewers have now commented on your article and suggested major revisions. When submitting the revised version of your article, it is better to clearly address all the reviews, comments and suggestions in terms of basic reporting, experimental design, validity of results and other comments.

Best wishes,

**Language Note:** The review process has identified that the English language must be improved. PeerJ can provide language editing services - please contact us at copyediting@peerj.com for pricing (be sure to provide your manuscript number and title). Alternatively, you should make your own arrangements to improve the language quality and provide details in your response letter. – PeerJ Staff

Reviewer 1 ·

Basic reporting

The authors have reflected all demands in the article.

Experimental design

The authors have reflected all demands in the article.

Validity of the findings

The authors have reflected all demands in the article.

Additional comments

The authors have reflected all demands in the article.

Reviewer 2 ·

Basic reporting

Well Written.

Experimental design

1、The experimental design is relatively simple, and there are few comparative methods. The author may consider adding some comparative methods.
Aslam M H, Zeeshan M O, Pedersoli M, et al. Privileged Knowledge Distillation for Dimensional Emotion Recognition in the Wild[C]//Proceedings of the IEEE/CVF Conference on Computer Vision and Pattern Recognition. 2023: 3337-3346.

2、Some ablation experiments or parameter analysis experiments can be supplemented, such as replacing the presentation layer or regression layer, to analyze the impact on emotion recognition.

Validity of the findings

1、The methods proposed by the author are all based on existing models. The core innovation of this article needs to be clearly stated in the Introduction and the methods in Chapter 3.

Reviewer 3 ·

Basic reporting

The authors have introduced a compelling study focusing on Internet-based cognitive behavioral therapy (iCBT) employing a transformer model for dimensional text-based emotion recognition. This model has been fine-tuned using publicly available datasets. Below are my comments and concerns regarding the manuscript:
1. The abstract does not sufficiently convey the technical contribution of the paper. It is recommended that the authors clarify their contributions in both the abstract and conclusion sections for better comprehension.
2. The paper lacks references to recent related work from reputable journals and conferences. Incorporating these references is essential to strengthen the paper's foundation in current research.

Experimental design

3. Further clarification is needed regarding the annotation process of the combined dataset, particularly on how the labels were assigned to the extended dataset.
4. The authors should provide more detail on the suitability of the dataset created in this manner and conduct a comparative analysis with existing related datasets to better justify its use.

Validity of the findings

5. The novelty of the proposed methods appears to be incremental. Additionally, the absence of visual representations, such as diagrams illustrating the proposed model, is notable and should be addressed.
6. The proposed methods should be compared with state-of-the-art techniques to validate their superiority. Without such comparisons, claiming the effectiveness of the proposed techniques is not justified.

·

Basic reporting

This article is fully within the scope of PeerJ computer Science. This paper presented a system for DL-based dimensional text-based emotion recognition for CA-based CBT. This method has good performance. the paper is well written and and easy to read, the organization is good.

Experimental design

1. Figures 2 and 3 seem a bit monotonous. It is recommended to improve the quality and aesthetics of the figures, which will help improve the quality of the paper.

2. In '3.4 Model Development', the description of the adapted ALBERT Model is not clear. This is not conducive to model reproduction. So it is recommended to provide a specific model structure Figure.

3. Some key details (e.g., experimental setup) are incomplete/unclear, or some key resources (e.g., proofs, code, data) are not furnished, without any further explanation or justification.

Validity of the findings

1.Quantitative comparative experiments support the results very well, but I recommend visualizing the losses during model training.

Reviewer 5 ·

Basic reporting

1. Clarity and Language: The paper must ensure clarity and professional language throughout. It's essential to go through the manuscript to refine the language for better readability and to ensure that it meets academic standards.
2. Literature Review: Expand the literature review to cover recent advancements and debates within the fields of emotion recognition, conversational agents, and their applications in cognitive behavioral therapy. This expansion should aim to position the study within the current state of research, highlighting its novelty and relevance.

Experimental design

1. Methodological Rigor: The methodology section requires more detailed descriptions of the experimental design, data collection, and analysis procedures. This would include specifying the model architectures, training details, and the rationale behind the chosen approaches. Ensuring reproducibility and transparency in research methods is crucial.
2. Validation of Findings: Strengthen the validity of the findings by providing a more comprehensive analysis of the results. This could involve additional statistical tests, comparisons with baseline models, or further exploration of the model's limitations and assumptions.

Validity of the findings

1. Discussion and Implications: The discussion section should more thoroughly explore the implications of the findings, especially in practical terms for cognitive behavioral therapy. It would benefit from a deeper analysis of how these findings contribute to existing knowledge and what they imply for future research or application.
2. Ethical Considerations: Given the application in therapeutic contexts, it's essential to address ethical considerations more thoroughly, including data privacy, potential biases in emotion recognition, and the implications of deploying such technology in sensitive settings.
3. Technical Details and Reproducibility: Provide more technical details regarding the models used, including hyperparameters, training procedures, and dataset specifics. Ensuring that other researchers can reproduce the findings is critical for the credibility of the research.
4. User Study and Usability Evaluation: The feasibility study with 20 participants provided valuable insights, but future work could benefit from larger, more diverse participant groups. Additionally, evaluating the system's usability and the perceived empathy of the conversational agent in more depth could provide more nuanced insights into its potential impact.
5. Addressing Limitations: Clearly articulate the study's limitations and how they might affect the generalizability of the findings. Discussing potential limitations openly will enhance the manuscript's credibility and provide a clearer path for future research.
6. Conclusion and Future Directions: The conclusion should succinctly summarize the main findings, their implications, and suggest clear directions for future research. Highlighting how this work advances the field and what questions remain unanswered can guide subsequent studies.

Additional comments

No comment.

---

## Round 0.2 · accepted · Accept

Dear authors,

Thank you for clearly addressing all the reviewers' comments. I confirm that the quality of your paper has improved. The paper is now ready for publication in light of the last revision.

Best wishes,

Reviewer 2 ·

Basic reporting

no comment.

Experimental design

no comment.

Validity of the findings

no comment.

·

Basic reporting

The author's response is reasonable, addressing my concern. I recommend accepting this manuscript.

Experimental design

good

Validity of the findings

The results are convincing

Reviewer 5 ·

Basic reporting

The author has made detailed revisions based on the feedback, and I believe that the revisions meet the innovative requirements of the paper.

Experimental design

no comment

Validity of the findings

no comment

Additional comments

no comment